# Economic evaluation of Health Extension Program packages in Ethiopia

**Lelisa Fekadu Assebe**[1,2,3]*, **Wondesen Nigatu Belete**[1,2,3], **Senait Alemayehu**[3,4],
**Elias Asfaw**[3,5], **Kora Tushune Godana**[6], **Yibeltal Kiflie Alemayehu**[3,6,7], **Alula M. Teklu**[3],
**Amanuel Yigezu**[3,4]

**1** Federal Ministry of Health, Addis Ababa, Ethiopia, **2** Department of Global Public Health and Primary Care
Medicine, University of Bergen, Bergen, Norway, **3** MERQ Consultancy PLC, Addis Ababa, Ethiopia,
**4** Ethiopian Public Health Institute, Addis Ababa, Ethiopia, **5** EAZ Consultancy and Research Service, Addis
Ababa, Ethiopia, **6** Department of Health Policy and Management, Jimma University, Jimma, Ethiopia,
**7** Department of Global Community Health and Behavioral Sciences, School of Public Health and Tropical
Medicine, Tulane University, New Orleans, LA, United States of America

* Lelfekadu1@gmail.com

GREECE

**Data Availability Statement:** All relevant data are
within the paper and its Supporting Information
files.

**Funding:** The study was conducted by MERQ
Consultancy PLC as part of a national assessment

## Abstract

### Background

Ethiopia launched the Health Extension Program (HEP) in 2004, aimed at ensuring equitable community-level healthcare services through Health Extension Workers. Despite the program's being a flagship initiative, there is limited evidence on whether investment in the program represents good value for money. This study assessed the cost and cost-effectiveness of HEP interventions to inform policy decisions for resource allocation and priority setting in Ethiopia.

### Methods

Twenty-one health care interventions were selected under the hygiene and sanitation, family health services, and disease prevention and control sub-domains. The ingredient bottom-up and top-down costing method was employed. Cost and cost-effectiveness were assessed from the provider perspective. Health outcomes were measured using life years gained (LYG). Incremental cost per LYG in relation to the gross domestic product (GDP) per capita of Ethiopia (US$852.80) was used to ascertain the cost-effectiveness. All costs were collected in Ethiopian birr and converted to United States dollars (US$) using the average exchange rate for 2018 (US$1 = 27.67 birr). Both costs and health outcomes were discounted by 3%.

### Result

The average unit cost of providing selected hygiene and sanitation, family health, and disease prevention and control services with the HEP was US$0.70, US$4.90, and US$7.40, respectively. The major cost driver was drugs and supplies, accounting for 53% and 68%, respectively, of the total cost. The average annual cost of delivering all the selected interventions was US$9,897. All interventions fall within 1 times GDP per capita per LYG, indicating

on the Health Extension Program of Ethiopia. The national assessment was financed through a grant from the Bill & Melinda Gates Foundation (INV-010174). The funder provided financial support to MERQ that covered professional fees for LFA, WNB, SA, EA, YKA, AMT, and AY, but did not have any additional role in the study design, data collection and analysis, decision to publish, or preparation of the manuscript. The specific roles of these authors are articulated in the "author contributions" section. EAZ Consultancy and Research Service did not provide any financial support for the study.

**Competing interests:** The authors declare that they have no competing interests. YKA and AMT are employees of MERQ Consultancy PLC. EA is an employee of EAZ Consultancy and Research Service. This does not alter our adherence to PLOS one policies on sharing data and materials.

**Abbreviations:** ANC, Antenatal care; CYP, Couple year of protection; DALY, disability-adjusted life year; DOTS, Directly observed treatment short course therapy; EDHS, Ethiopian Demographic and Health Survey; FMOH, Federal Ministry of Health; GDP, Gross domestic product; HEP, Health Extension Program; HEWs, Health Extension Workers; HIV, Human Immunodeficiency Virus; ICER, Incremental cost effectiveness ratio; LiST, Lives saved tool; M and E, Monitoring and Evaluation; IRS, Indoor residual spray; ITN, Insecticide treated bed nets; LYG, Life years gained; OCP, oral contraceptive pills; ORS, oral rehydration salt; TB, Tuberculosis; US$, United States dollars; TT, tetanus toxoid.

that they are very cost-effective (ranges: US$22–$295 per LYG). Overall, the HEP is cost-effective by investing US$77.40 for every LYG.

## Conclusion

The unit cost estimates of HEP interventions are crucial for priority-setting, resource mobilization, and program planning. This study found that the program is very cost-effective in delivering community health services.

## Introduction

Ethiopia has a three-tier healthcare delivery system, with primary, secondary, and tertiary level units. The primary health care unit is the lowest level of the tier system, comprising primary hospitals, health centers, and their satellite health posts, all linked by a referral network. Health posts are the most peripheral units, providing mainly preventive care and selected curative services [1, 2]. Over the past years there was a steady increase in the number of health posts, from 6,191 in 2013 to 17,187 in 2017 [3].

The Health Extension Program (HEP) is one of the most innovative government-led community-based health programs in Ethiopia. It was introduced in 2004 with the primary objective of achieving universal health coverage (UHC) through the provision of equitable health care to the community within a context of limited resources. The HEP package was updated in 2011 with more interventions. The HEP has five main components comprising 16 packages: disease prevention and control (DPC), family health, hygiene and environmental sanitation, health education and communication, and first aid. Health Extension Workers (HEWs) provide the health service packages through three modalities: static (e.g., at a health post), outreach (e.g., immunization campaigns), and home-to-home visits [4]. After a year of pre-service training, HEWs are deployed in each district or kebele (the lowest administrative unit in Ethiopia) to deliver the HEP services [5].

Ethiopia has made substantial improvements in health outcomes: for example, the maternal mortality ratio (MMR) decreased from 676 deaths per 100,000 in 2011 to 401 in 2017. Under-five mortality per 1,000 live births decreased from 123 in 2005 to 59 in 2019 leading to an average life expectancy at birth of 65.5 years. HIV, Tuberculosis (TB), and malaria-related morbidity and mortality have significantly decreased in the last decades [2]. HEP's contribution in the areas of family planning, maternal care, and reducing morbidity and mortality related to pneumonia, diarrhea, and malaria has been substantial. In addition, the improvement in the potential coverage of health service to more than 90% in 2019 was primarily attributed to the extensive investment of the country in the primary health care unit initiatives that would otherwise not be achievable [2, 6, 7].

Although the HEP is a flagship program in Ethiopia, there is limited evidence available for whether the investment in the HEP provides good value for money. A study on the cost-effectiveness of the community-based health program (focused on reproductive, maternal, newborn and child health interventions) in Ethiopia found that the incremental cost effectiveness ratio (ICER) of HEP was US$ 999 per life years gained (LYG), which was deemed cost-effective based on the WHO CHOICE threshold of Ethiopia's gross domestic product [8].

Previous cost-effectiveness studies on the HEP packages are very few in number and lack a detailed evaluation of the various HEP components. Therefore, a broader assessment of health impacts beyond specific packages would capture the positive contribution of community-based programs in different areas of health services. The aim of this study is to estimate the

cost and cost-effectiveness of selected HEP interventions to assist decision-makers and program managers in identifying interventions representing the best value for money as well as in prioritizing the allocation of scarce resources.

## Methods

### Study setting

The study was conducted as part of the Ethiopian HEP's national assessment of the relevance of the HEP packages and the provision of services [9]. Both primary and secondary data were used. Primary data were collected from 300 health posts, 54 health centers and 57 district health offices from June 7 to July 1, 2019.

### Interventions

The following 21 interventions were selected based on the availability of cost data, effectiveness measures, and interventions, mostly implemented by HEWs (Table 1).

### Costing approaches

Costing was conducted from a provider perspective, where the costs incurred by the government were included [10]. The bottom-up ingredient and top-down costing methods was

**Table 1. Selected HEP interventions from the family health services, disease prevention and control, hygiene and environmental sanitation subdomains.**

| |
|---|
| **Family health services (preventive)** |
| 1.1 Maternal and child health |
| Antenatal care |
| Tetanus toxoid vaccinations |
| Iron folate supplementation |
| Family planning: oral contraceptive pills (OCP), condoms, injectables and implants |
| 1.2 Expanded program of immunization (EPI) |
| Pentavalent vaccination |
| Measles vaccination |
| Pneumococcal vaccination |
| **Disease prevention and control (curative services)** |
| 2.1 Human immunodeficiency virus (HIV) testing and counselling |
| HIV testing and counselling |
| 2.2 TB prevention and control |
| Directly observed treatment short course therapy (DOTS) |
| 2.3 Malaria prevention and control |
| Insecticide treated bed nets (ITN) |
| Indoor residual spray (IRS) |
| Malaria treatment |
| 2.4 Diarrheal disease management (oral rehydration salt (ORS) and Zinc) |
| 2.5 Pneumonia treatment (i.e. cotrimoxazole, amoxicillin, gentamycin) |
| **Hygiene and environmental sanitation** |
| 3.1 Improved water source |
| 3.2 Hand washing with soup |
| 3.3 Hygienic disposal of children's stool |
| 3.4 Latrine use |

applied. The bottom-up ingredient costing method specifies activities under each intervention, measures the resources used for the activities (i.e., expenditure), and assigns prices to the resources. The top-down costing method considers overall spending at a central level to allocate some of the costs to each intervention [11]. All cost inputs from both methods were cross-checked and accounted for to avoid double-counting. The cost items that last for more than one year were treated as capital and those used within a year were considered recurrent. The capital inputs include pre-service and in-service training, the building of health posts, and equipment. The recurrent inputs include supplies, personnel, supervision, and review meetings.

## Cost inputs and analysis

The cost components include personnel, medicine, supplies, infrastructure, capacity building, and equipment (Table 2). To estimate personnel costs, the number and salaries of HEWs with level 3 and 4 career certifications were obtained [12]. The average HEW salary per minute was then calculated based on 8 working hours per day and 22 working days per month. The average HEW salary per minute was multiplied by the average working time HEWs spent on each service (i.e., a time motion study) to obtain personnel costs per service [13]. The total supply costs per service include the cost of items, the pharmaceutical management supply cost, and vaccination wastage. The cost of supervision and review meetings at the national and subnational levels was estimated per health post per year. Health posts are made from hollow brick, stone, and wood, with an average service life of 30, 20, and 15 years, respectively. The average cost of the three types of buildings was calculated and subsequently annualized. Similarly, 5 life-years for equipment, 10 for pre-service training, and 5 for in-service training were considered. The capital and recurrent costs (except personnel) were distributed to each service using the total number of people served per health post as an allocation base. The cost per intervention was then estimated by adding the unit costs to the recurrent and capital costs.

To estimate costs per client for services requiring more than one visit, the number of visits was multiplied by the cost per service. Similarly, the cost per couple year of protection (CYP) was estimated for family-planning services. We then estimated the total annual cost of delivering selected HEP interventions at health posts by multiplying the unit cost of selected HEP

**Table 2. Description of input costs and source of data.**

| Costs inputs | Description | Source of Data |
|---|---|---|
| Building | Average cost of health post made of hollow block, wood/mud and stone. | Primary data |
| Equipment | Number of functional equipment per health post. | Primary data |
| Pre-service training | Cost of pre-service training per year. | Federal Ministry of Health (FMOH) report |
| In-service training | Cost of providing integrated refresher training. | FMOH report |
| Personnel | Personnel time per client served estimated by time motion study and total client served. | Primary data [12, 13] |
| Supplies* | Supplies used for provision of selected service. | Primary data |
| Supervision | Number of supervisions conducted at all level per year. | Primary data, FMOH report, and expert opinion. |
| Review meeting | Number of review meetings held by health centers and health offices with the HEWs per year. | Primary data |

* Supplies includes medicines, consumables, and stationery.

interventions by the number of clients served. All costs were collected in Ethiopian birr and converted to United States dollars (US$) using the average exchange rate for 2018 (US $1 = 27.67 birr). Costs were inflated to 2018 rates using the Ethiopian consumer price index. A discount rate of 3% was applied [14, 15]. The cost analysis was carried out on an Excel spreadsheet [16].

## Health outcome measures

Health outcomes were measured in LYG and estimated using Spectrum software. Three steps were used to calculate the number of LYG due to the HEP to estimate the number of lives saved by an intervention at a given coverage. First, the current health services coverage, including the HEP, was extracted from the 2016 Ethiopian Demographic and Health Survey (EDHS; i.e., the health system with HEP). Second, by using scientific literature on impact studies, we created counterfactual scenarios of what would have happened to the health system coverage in 2016 were there is no HEP (i.e., a health system without the HEP). Baseline coverage was presumed to be similar for both steps and was extracted from the 2005 and 2011 EDHS [17, 18]; linear interpolation was then made between baseline and current coverage for the two steps (Table 3). Spectrum software [19] (lives saved tool [LiST], FamPlan, and TB impact and

**Table 3. Coverage inputs for effectiveness measures.**

| Intervention | Target population | Baseline coverage (%) (2005) | With HEP coverage (%) (2016) | Without HEP coverage (%) (2016) | Source |
|---|---|---|---|---|---|
| Ante-natal care (ANC):provision of iron, pregnant mother malaria treatment and tetanus toxoid (TT) vaccination | Total number of estimated pregnancies/births | 28 | 62 | 42 | [17, 22, 23] |
| Percentage receiving two or more TT injections during the pregnancy for the most recent live birth. | Total number of estimated pregnancies/births | 32 | 41 | 34 | [8, 17, 22] |
| Iron supplementation | Total number of estimated pregnancies/births | 10 | 42 | 24 | [17, 22, 23] |
| Family planning | Non-pregnant | 24 | 35 | 24 | [17, 22] |
| EPI: Pentavalent 3 | Estimated live births | 29 | 53 | 30 | [7, 17, 22] |
| EPI: Measles | Estimated live births | 28.5 | 54 | 33 | [7, 17, 22] |
| TB: DOTs* | Number of populations in need of TB treatment | 84 | 89 | 83 | [24–26] |
| Malaria: ITN | Population at risk (60%) of malaria | 33 | 40 | 32 | [6, 27, 28] |
| Malaria: IRS | Population residing in highland fringe/epidemic-prone areas | 14.2 | 29 | 21 | [6, 27, 28] |
| Malaria-diagnosis and treatment* | Prevalence of malaria in all age group | 24.2 | 38 | 18 | [18, 22, 27] |
| Diarrheal treatment (zinc)* | Diarrhoea prevalence among children <5 years | 0.3 | 33 | 28 | [6, 18, 22], primary data |
| Diarrheal treatment (ORS)* | | 40 | 46 | 40 | [6, 18, 22] |
| Acute respiratory tract infection and Pneumonia treatment*¥ | Pneumonia Prevalence among children <5 years | 13 | 31 | 22 | [18, 22, 29] |
| Improved water source | Households | 51 | 62 | 58.9 | [17, 22] |
| Hand washing with soap | Households | 50 | 60 | 57.2 | [17, 22] |

*Baseline year is 2010.

¥ Treatment includes the provision of cotrimoxazole, amoxicillin, and gentamycin.

modelling estimate [TIME]) then generated the number of lives saved for both steps. As a third step, the difference in the number of lives saved between the health system coverage (Health system with HEP) and the counterfactual (health system without the HEP) was calculated as the number of lives saved due to the HEP.

The mathematical formula to estimate the number of lives saved due to an intervention is:

$$LS = cause\ specific\ mortality * intervention\ coverage\ change * AF * E \tag{1}$$

$$\text{Where}\quad LS: \text{lives saved;}$$

$$AF: \text{affected fraction; and}$$

$$E: \text{effectiveness of the intervention.}$$

The number of lives saved was then multiplied by remaining life expectancy retrieved from Ethiopia's life tables [20] to obtain the number of LYG [21]. Remaining life expectancy was calculated by weighing the number of male and female children below five years old and averaging the remaining years of life. Likewise, the weighted average life expectancy is calculated for women of reproductive age.

The life years gained were calculated as:

$$LYG = LS * average\ remaining\ life\ expectancy \tag{2}$$

$$\text{Where } LYG: \text{life}-\text{years gained; and}$$

$$LS: \text{lives saved.}$$

The Lives Saved Tool is a mathematical modelling tool used to estimate the effect of changes in coverage on mortality in low-and middle-income countries. The FamPlan model estimates the number of lives saved by considering family planning needs, the mix of methods, the contraceptive prevalence rate, and other criteria. TIME is an epidemiological compartmental transmission model that projects the drug-susceptible and multidrug-resistant TB burden.

## Cost-effectiveness analysis

Since the founding of the HEP in 2004, the health system has incurred additional costs of health services. This additional cost to the health system will bring additional benefits to the community. We calculated the total incremental cost of a health service intervention due to the HEP by multiplying the additional number of the population covered due to the HEP and the unit cost of the selected intervention. The incremental cost-effectiveness ratio (ICER) is calculated as the ratio of the additional costs of intervention divided by the additional benefit of the intervention due to the HEP. The ICER is calculated for each intervention as well as for the whole selected HEP packages.

$$ICER = \frac{\triangle cost}{\triangle Effectiveness} \tag{3}$$

$$\text{Where } ICER: \text{Incremental Cost}-\text{effectiveness ratio;}$$

$$\triangle\text{Cost}: \text{incremental cost due to the HEP; and}$$

$$\triangle\text{Effectiveness}: \text{incremental effectiveness due to the HEP.}$$

A one-year time horizon was applied. All cost and health outcome measures were discounted at 3%. The WHO's CHOICE recommendation based on the GDP per capita for the year 2018 (i.e., US$852.80) was used to determine cost-effectiveness [27].

### Sensitivity analysis

One-way sensitivity analysis was performed by varying the unit costs, the discount rate, the life-time duration of equipment, buildings, pre-service and in-service training, and the salary of HEWs and others to check for the robustness of the findings when those parameters are changed.

## Results

### Cost analysis

The unit cost was US$4.90 for family health service, US$7.40 for DPC, and US$0.70 for the hygiene and sanitation component of the HEP. The cost of delivering family health service by HEWs ranged from US$0.30 (iron folate supplementation) to US$9.90 (family planning: implant). Likewise, it ranged from US$2.0 for diarrheal disease management to US$43.10 for TB diagnosis, a service that is part of the DPC component. The unit costs of hygiene and environmental sanitation–which include improved water sources, hand-washing with soap, the hygienic disposal of children's stool, and latrine use–range from US$0.60 to US$0.80 (Table 4). In addition, the average annual cost of delivering the selected HEP interventions per health post is US$9,897.

### Unit cost disaggregated by HEP components

The major cost drivers of the cost of family health services were drugs and supplies, which accounted for 68% of the total, followed by equipment (15%), capacity-building (8%), construction (8%), and personnel (4%) (Fig 1).

**Table 4. Unit cost of selected HEP interventions by service delivery modalities in Ethiopia, 2018.**

| Package | Service Type | unit cost (US $) |
|---|---|---|
| Hygiene and environmental sanitation | Improved water source | 0.6 |
| | Hand washing with soap | 0.7 |
| | Hygienic disposal of children's stool | 0.7 |
| | Use of latrine | 0.8 |
| | **Average cost** | **0.7** |
| Family health service | ANC | 1.88 |
| | Family planning: OCP | 1.2 |
| | Condom | 1.8 |
| | Injectable | 3.2 |
| | Family planning -Implant | 10.5 |
| | Pentavalent vaccination | 15.2 |
| | Measles vaccination | 2.5 |
| | Tetanus toxoid vaccination | 3.0 |
| | Iron folate supplementation | 0.7 |
| | Pneumococcal vaccination | 9.4 |
| | **Average cost** | **4.9** |
| DPC | HIV testing and counselling | 3.7 |
| | TB treatment (DOTS) | 43.1 |
| | Malaria case management | 1.8 |
| | Malaria prevention: ITN | 2.1 |
| | IRS | 3.4 |
| | Diarrheal disease | 2.0 |
| | Pneumonia treatment | 2.6 |
| | **Average cost** | **7.4** |

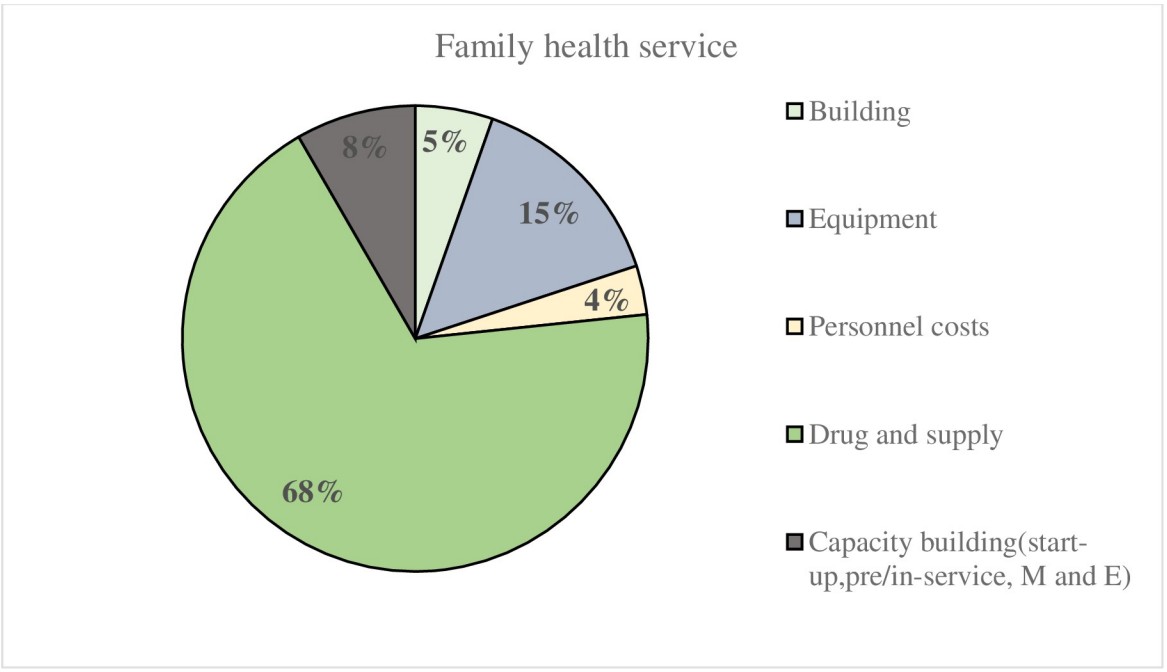

**Fig 1. Percentage distribution of family health services unit costs by ingredients.**

The study also showed that drugs and supplies were the leading cost drivers of DPC interventions (53%) (Fig 2). The cost of personnel (15%) followed by building (14%), equipment (6%), and capacity building (12%) also contributed to the unit cost of DPC interventions.

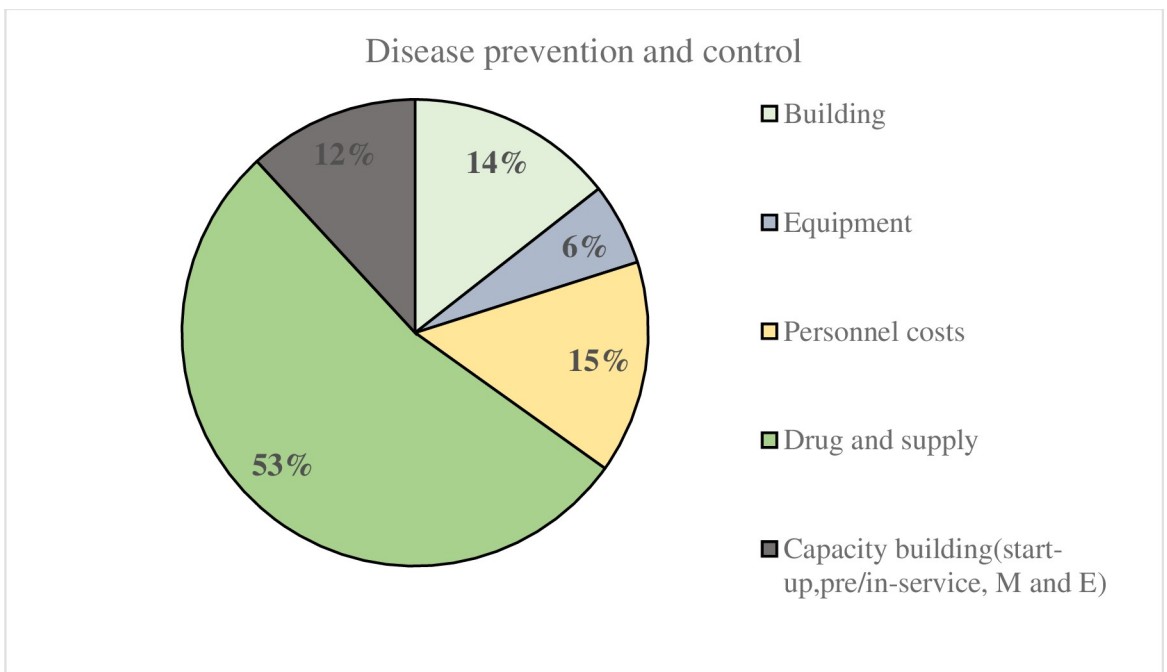

**Fig 2. Percentage distribution of DPC unit costs by ingredient.**

Personnel and capacity-building costs were the only cost centers for the hygiene and environmental sanitation package. Personnel costs accounted for 18%, 24%, 24%, and 32% of the costs of improved water sources, hand washing with soap, the hygienic disposal of children's stools, and the use of latrines, respectively. The remaining share of the expenditure was contributed by capacity building, including start-up and monitoring and evaluation.

## Cost-effectiveness analysis

The cost-effectiveness results, ranked from the most cost-effective intervention to the least are presented in Fig 3. The cost-effectiveness ratios of improved water sources, measles vaccination, hand-washing with soap, TT vaccination for pregnant women, antenatal care and iron supplementation for pregnant women, pentavalent vaccination, and oral antibiotics for pneumonia treatment were below the overall average and range between US$21.60 and US$67.20 per LYG. This suggests that providing the above interventions through the HEP would incur an additional US$21.60 to US$67.20 to the health care system for each additional LYG. Similarly, the provision of zinc and ORS for diarrheal case management, malaria case management, pneumococcal vaccination, TB treatment follow-up (i.e., DOTs), long-lasting insecticide-treated net, and family planning services had ICERs that ranged between US$78.10 and US$295.40 per LYG. Overall, the incremental costs of the HEP to the health system were US$24,886,899, with an additional benefit of 321,463.0 LYG over 15 years. In summary, for each additional LYG, the cost-effectiveness ratio of the HEP is US$ 77.40.

All the interventions are considered very cos-effective, as the cost per LYG is less than 1 times Ethiopian GDP per capita. Among the interventions, the ICER for improved water

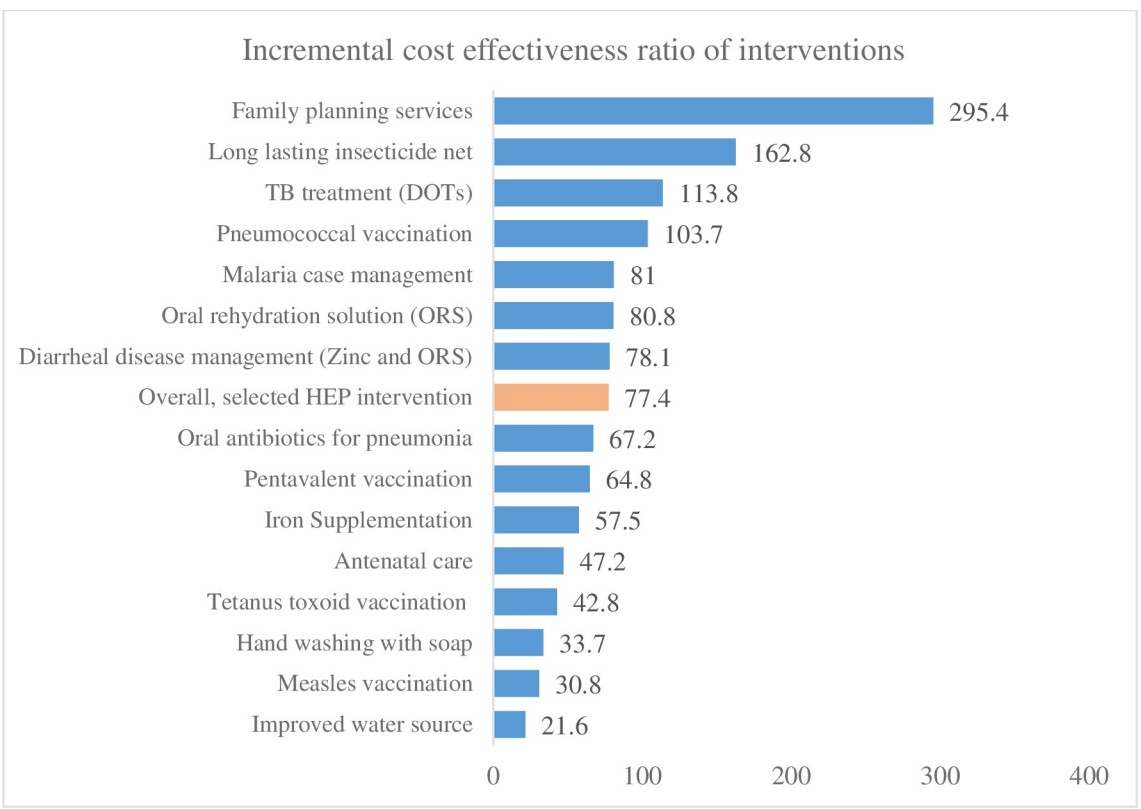

**Fig 3. Incremental cost-effectiveness ratio of selected HEP interventions in Ethiopia, 2018.**

source, measles vaccination, hand-washing with soap, TT vaccination for pregnant women, antenatal care, iron supplementation for pregnant women pentavalent vaccination, and oral antibiotics for pneumonia treatment less than 10% of the country's GDP per capita for each additional LYG. Interventions like ORS for diarrheal disease control, zinc, and ORS for diarrheal case management and the treatment of malaria, pneumococcal vaccination, TB treatment follow up (DOTs), LLIN use, and the provision of family planning service have costs ranging from 10% to 34% of the country's GDP per capita for each LYG (Fig 3).

## Sensitivity analysis

Figs 4 and 5 present the results of the sensitivity analysis, which was conducted by varying the various parameters, such as unit cost, discounting LYG, life-years of capital items, pre-service and in-service training, and salary scale. Among all the interventions, discounting the LYG had the strong effect impact on the base case ICER, which changed slightly, when the discount rate for LYG was altered from no discounting to discounting by 6%. In almost all interventions, age discounting increased the base case ICER by a factor of two when changed from 3% to 6%. Conversely, when it was changed from a 3% discount to no discounting, the base case ICER fell by half. Life-years of equipment had the strongest effect on ICER after discounting the LYG. For example, when equipment life-years was changed from five to three, the ICER of the TT vaccination increased by 30% (Fig 4). Conversely, when equipment life-years were increased from five to seven, the ICER fell by 10%. The change in the ICER of the TT vaccination is a good example of the effect of poor-quality equipment and furniture, which incur a 30% additional cost to gain one more life year due to their decreased number of service years. Other variables, such as unit costs, service coverage, discounting costs, the total number of service users, building service years, building costs, HEWs'

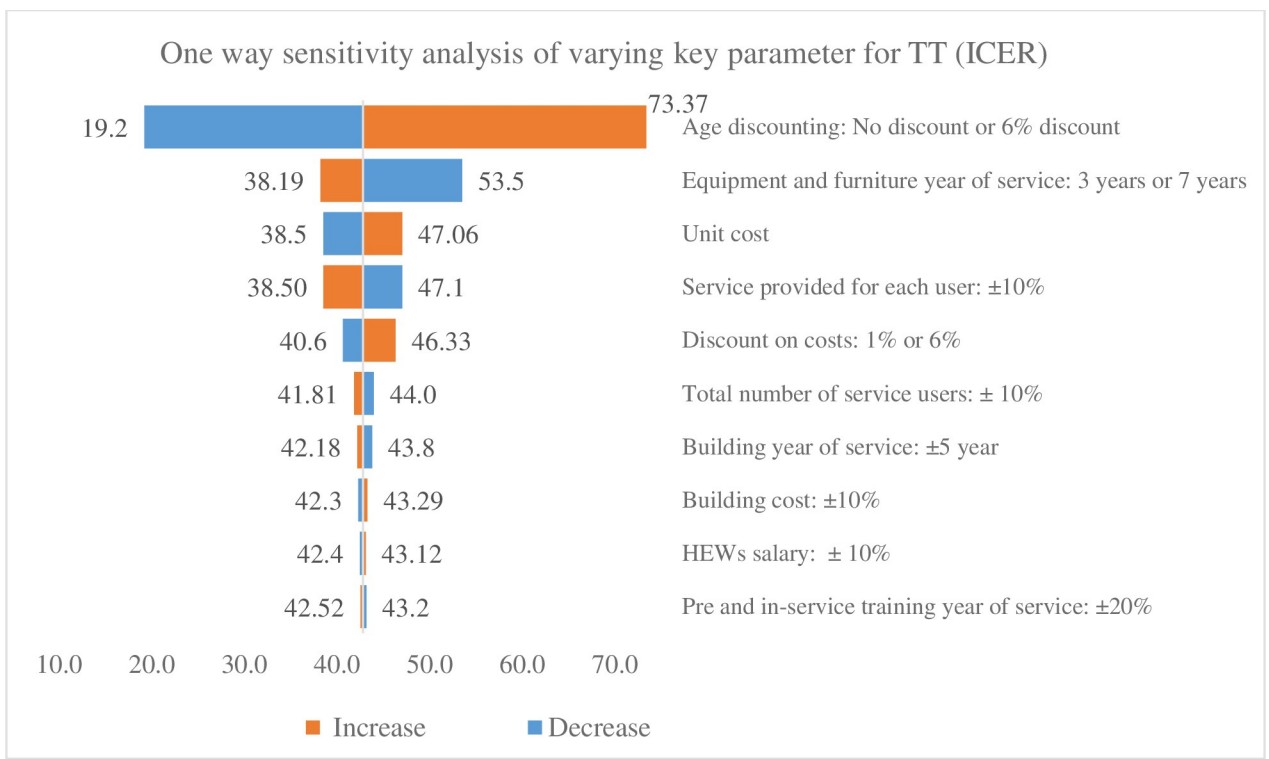

**Fig 4. One-way sensitivity analysis showing cost-effectiveness ratio of TT injections over a range of key parameters.**

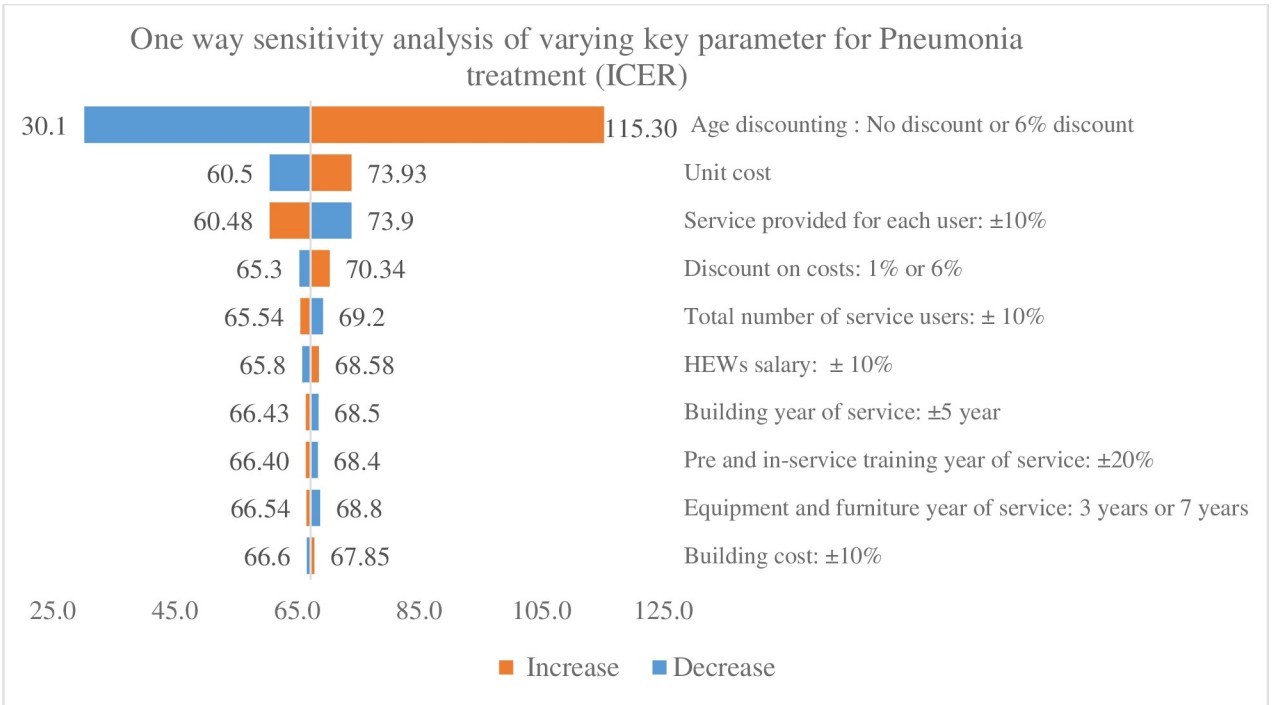

**Fig 5. One-way sensitivity analysis showing cost-effectiveness ratio of pneumonia treatment over a range of key parameters.**

salaries, and pre-service and in-service training years of service, have a minimal effect on the baseline ICER. Although changes in the rate used for age discounting do affect the ICER for the lower (19.2) and upper bounds (73.3), the range still lies within 1 times GDP per capita of the country, and the overall finding is robust with the base case results.

## Discussion

This study has estimated the cost and cost-effectiveness of providing selected HEP packages. The unit costs of providing the HEP services ranges from US$0.70 (iron folate supplementation) to US$15.0 (pentavalent vaccination) for family health services. Similarly, the unit cost of providing DPC services ranges from US$2.0 (diarrheal disease management) to US$43.10 (TB DOT). The incremental cost per LYG was 77.4%. The unit costs and cost-effectiveness of the HEP based on health need and supply constraints provide contextual evidence for decision-makers to prioritize healthcare interventions and allocate resources efficiently in order to improve population health.

The weighted mean cost of family health service and DPC services is US$4.90 and US$7.40, respectively. The cost of DPC interventions has a higher variability in their unit cost than family health services costs. A 2015 study conducted in Ghana estimated the unit cost of providing health care at the community level at US$5.10, with curative services accounting for 34% of the unit cost and preventive services for 56%. Although the total costs depend on the interventions included in the community health program, the estimates for family health and DPC services are similar to the average cost estimated in Ghana [30]. The costs of providing diarrheal case management, oral antibiotics for pneumonia and malaria treatment through the HEP are US$2.0, US$2.60, and US$1.80 respectively. Other studies carried out in seven countries indicated that the cost per diagnosis ranged from US$2.44 to US$13.71 for case management of diarrhea, US$2.17 to US$17.54 for malaria, and US$1.70 to US$12.94 for pneumonia treatment between

2010 and 2012 [31]. Another study, in Zambia, estimated the unit cost of uncomplicated malaria treatment at home to be US$4.22 [32]. The variation in the cost of providing the services might be attributed to the costing method, the resources included, and differences in how resources are combined for the services, the prevalence of the disease, and the countries' economies. The unit cost of providing IRS through the HEP was US$3.4. A 2013 study conducted in Ethiopia estimated that district-based IRS would cost more than community-based IRS, both in terms of cost per district and in terms of cost per person protected [33]. Drugs and supplies were the key cost drivers of the program, unlike in the previous study in Ghana, where staff costs accounted for the largest proportion of unit costs. This may be attributed to the difference in the average number of workers per health facility, the difference in the estimation of staff time, or the variation in the wage scale between the two studies [30].

In this study, the average annualized cost of providing services through the HEP was US$9,897 marginally lower than in the study conducted in Ghana, which may be attributed to the high annualized cost of personnel in the latter and differences in delivery mechanisms, and the cost structure of the two countries [30]. Another previous study, conducted in India in 2014, estimated the annual cost of delivering healthcare interventions through community health workers to be US$19,381 [34].

In this study, the HEP was shown to be very cost-effective, costing US$77.40 for every LYG. Thus, the HEP leads to a higher proportion of lives saved and a more cost-effective approach to delivering essential health services to rural and vulnerable communities, where access to qualified staff is limited. Similarly, previous studies and systematic reviews of the cost-effectiveness of community health workers have indicated that providing essential healthcare services through community health workers is a cost-effective or very cost-effective strategy [35, 36]. One study conducted in rural Nepal also estimated the ICER to be US$211 per LYG to provide healthcare interventions through women's group. Similarly, other community health programs administered through participatory women's groups estimated an ICER of US$79 per disability-adjusted life year (DALY) averted to provide healthcare interventions in rural Malawi that benefitted the community with profound maternal and child health gains [37]. A 2013 study conducted in selected districts in Ethiopia, Kenya and Indonesia indicated that community health program/HEP model is cost-effective, with an ICER of US$999 per LYG in Ethiopia, US$82 per LYG in Kenya, and US$3,397 per LYG in Indonesia. Our study, however, found that the HEP is a very cost-effective program. In this study, the difference in ICER (i.e. US$999 vs 77.40 per LYG) is attributed to the differences in cost estimation, where we used national estimates and considered building costs that were not addressed in the previous study. Moreover, the implementation period of the HEP that we used is longer than the previous study (i.e., 3 years vs 8 and 13 years) (8). In particular, the cost-effectiveness of providing treatment for malaria through HEWs is US$81 per LYG. In Ghana, this was US$90 per DALY averted in 2012, a very cost-effective investment in both countries [38]. Our study showed measles vaccination through community health workers to be very cost-effective in Ethiopia, with an ICER of US$30.80 per LYG. The finding is similar to that of a study carried out in India, which reported US$162 per DALY averted [39].

Another study conducted on community health workers estimated the ICER for key child survival interventions to cost US$67 (range: 27–92) per DALY averted in Mozambique, Rwanda and Malawi [40]. The study indicated that community health programs represent an attractive and low-cost investment that increases the coverage of key child interventions and decreases child mortality. A previous study conducted in Bangladesh compared community healthcare with home-based care for maternal and neonatal interventions. It found that implementation of the home-care strategy was very cost-effective with an ICER of US$ 103 per DALY averted [41], indicating that the provision of maternal and child healthcare

interventions through home-to-home visits is very cost-effective. Although it is difficult to make a realistic comparison of ICER data due to the variability of time horizons, settings, perspectives of the studies, the disease burden and range of interventions, aggregate evidence from other studies provide insights and the genuine lesson that community health programs are a cost-effective or a very cost-effective strategy. The provision of health care services through the HEP, however, is not a standalone strategy, but a complementary approach to other mechanisms of delivering healthcare services in the country.

The following limitations apply to this study. First, although this study considered a wide range of HEP interventions, it excluded interventions with an absence of clear cost and outcome measures; this could affect the findings of this study. Second, the health outcome measure takes into account only the mortality aspect of the intervention, and the ICER reported in this study may be overestimated because it ignores morbidity effects. Third, the estimation of the number of lives saved due to limitations in Spectrum's LiST method does not reflect the effects of malaria and pneumonia on adult mortality, so our estimates are likely to have underestimated the actual number of lives saved by these interventions. Fourth, by their very nature, community-based programs function outside the formal systems, and the provider perspective does not capture all of the social costs associated with the HEP; incorporating other perspectives may yield better outcomes. Lastly, the literature's estimate of the effectiveness or health outcome data was not nationally representative and, included short evaluation periods that are further affected by the availability and quality data, which influences the HEP's effect. Notwithstanding a number of methodological and data-availability limitations, the study provided cost and cost-effectiveness estimates for national HEP programs and illustrated the potential effect of the program in Ethiopia. Overall, the findings of this study represent an additional contribution to the wider (but still limited) literature that suggests that HEP strategies tend to be cost-effective and improve the coverage of essential services. Potential research focusing on implementation modalities, how the HEP can affect the wider health system, and what broader social costs and benefits they can offer is important.

## Conclusion

The costing of HEP interventions is important for setting priorities, mobilizing resources, and advocacy, as well as for various program-planning and budgeting activities. All of the selected interventions were found to be very cost-effective. The cost-effectiveness analysis will enhance the case for stronger HEP investment and can be used during priority setting to identify the most cost-effective packages of interventions.

## Supporting information

**S1 Dataset.**
(XLSX)

## Acknowledgments

The authors would like to thank Professor David Hotchkiss from the school of Public Health and Tropical Medicine at Tulane University, Professor Stéphane Verguet from the Harvard T. H. Chan School of Public Health, and Dr. Nejmudin Kedir Bilal from Johns Hopkins Bloomberg School of Public Health for valuable comments on an earlier version of the manuscript. We thank Dr. Girmay Medhin and Dr. Girma Azene for useful discussions. The authors also thank Ethiopia's Ministry of Health for making this study possible and acknowledge all the study facilities involved in this study.

## Author Contributions

**Conceptualization:** Lelisa Fekadu Assebe, Kora Tushune Godana, Yibeltal Kiflie Alemayehu, Alula M. Teklu.

**Data curation:** Lelisa Fekadu Assebe, Wondesen Nigatu Belete, Senait Alemayehu, Elias Asfaw, Amanuel Yigezu.

**Formal analysis:** Lelisa Fekadu Assebe, Wondesen Nigatu Belete, Senait Alemayehu, Elias Asfaw, Yibeltal Kiflie Alemayehu, Amanuel Yigezu.

**Funding acquisition:** Alula M. Teklu.

**Methodology:** Lelisa Fekadu Assebe, Amanuel Yigezu.

**Resources:** Alula M. Teklu.

**Supervision:** Elias Asfaw.

**Writing – original draft:** Lelisa Fekadu Assebe, Amanuel Yigezu.

**Writing – review & editing:** Lelisa Fekadu Assebe, Wondesen Nigatu Belete, Senait Alemayehu, Elias Asfaw, Kora Tushune Godana, Yibeltal Kiflie Alemayehu, Alula M. Teklu, Amanuel Yigezu.

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
