## [Decision Letter · Decision Letter 0]

7 Dec 2020

PONE-D-20-27778

Economic evaluation of Health Extension Program packages in Ethiopia

PLOS ONE

Dear Dr. Assebe,

Thank you for submitting your manuscript to PLOS ONE. After careful consideration, we feel that it has merit but does not fully meet PLOS ONE’s publication criteria as it currently stands. Therefore, we invite you to submit a revised version of the manuscript that addresses the points raised during the review process.

During the revision process of your paper, please take into careful consideration the reviewers' comments. In addition, please re-check the manuscript for the grammatical errors that appear throughout the manuscript and make the appropriate corrections.

We look forward to receiving your revised manuscript.

Kind regards,

Athina Economou

Academic Editor

PLOS ONE

Journal Requirements:

2.) We note that you have indicated that data from this study are available upon request. PLOS only allows data to be available upon request if there are legal or ethical restrictions on sharing data publicly. For information on unacceptable data access restrictions, please see http://journals.plos.org/plosone/s/data-availability#loc-unacceptable-data-access-restrictions.

3.) Please amend your list of authors on the manuscript to ensure that each author is linked to an affiliation. Authors’ affiliations should reflect the institution where the work was done (if authors moved subsequently, you can also list the new affiliation stating “current affiliation:….” as necessary).

4.) Thank you for stating the following in the Financial Disclosure section:

'Bill & Melinda Gates Foundation funded the study. The funder had no role in the conception, data collection, analysis and write up of this study.'

We note that one or more of the authors are employed by a commercial company: Abenezer Consulting and Research PLC, and Monitoring, Evaluation, Research and Quality Improvement (MERQ) consultancy.

5.) We note you have included a table to which you do not refer in the text of your manuscript. Please ensure that you refer to Table 2 and 4 in your text; if accepted, production will need this reference to link the reader to the Table.

Reviewers' comments:

Reviewer's Responses to Questions

**Comments to the Author**

1. Is the manuscript technically sound, and do the data support the conclusions?

Reviewer #1: Yes

Reviewer #2: Yes

2. Has the statistical analysis been performed appropriately and rigorously? 

Reviewer #1: Yes

Reviewer #2: Yes

3. Have the authors made all data underlying the findings in their manuscript fully available?

Reviewer #1: Yes

Reviewer #2: Yes

4. Is the manuscript presented in an intelligible fashion and written in standard English?

Reviewer #1: Yes

Reviewer #2: Yes

5. Review Comments to the Author

Reviewer #1: The manuscript is well written and easy to follow. However, taking the following into consideration will improve the manuscript:

In the introduction, you mention that the country has made substantial improvements in health outcomes. Provide a brief synopsis of the statistics to show those improvements.

Reviewer #2: The manuscript is technically sound except for a few issues. In your costing approach, there was no mention of the possibility of double counting costs using the two approaches and how you handled double counting if it happened. If no double counting happened please mention it.

Please explain why the 2018 inflationary and exchange rate was used for costs data collected in 2019.

In table 2, please align supervision and its description properly.

Please explain the data source for the number of people/clients served. Was this collected on a daily bases during data collection period, or the number was taken from secondary data source e,g hospital records during set period of time.

Please indicate what software was used to make the linear interpolation between baseline and current coverage before the Spectrum analysis.

Personnel costs contribution for hygiene and environment has 3 percentage points but 4 items mentioned; utilization of latrines seems to have no percentage point mentioned

major part of the discussion was comparing study results with other studies and little on impact of these results on current policy.

Small grammatical errors throughout the manuscript that need revision.

6. PLOS authors have the option to publish the peer review history of their article (what does this mean?). If published, this will include your full peer review and any attached files.

Reviewer #1: **Yes: **Naomi Setshegetso

Reviewer #2: No

---

## [Author Response · Author response to Decision Letter 0]

18 Dec 2020

Dec 16, 2020 

To: Athina Economou and/or other editor(s):

Journal: PLOS ONE

Dear Editors,

Thank you for allowing us to submit a revision of our paper “Economic evaluation of Health Extension Program packages in Ethiopia” (PONE-D-20-27778). We thank the reviewers for their constructive suggestions and valuable comments. A point-by-point response to the reviewer’s and editors comments can be found below. All page numbers apply to the “Main document clean version” document.

Thank you very much for your consideration of our manuscript. We look forward to hearing from you again. 

Best wishes,

Lelisa Fekadu on behalf of all authors

 

Reviewer 1 

1. The manuscript is well written and easy to follow. However, considering the following will improve the manuscript: In the introduction, you mention that the country has made substantial improvements in health outcomes. Provide a brief synopsis of the statistics to show those improvements.

Reply: Thank you for the comment and we added the sentence in page 4 and line number 88-92. “Ethiopia has made substantial improvements in health outcomes: for example, the maternal mortality ratio (MMR) decreased from 676 deaths per 100,000 in 2011 to 401 in 2017. Under-five mortality per 1,000 live births decreased from 123 in 2005 to 59 in 2019 leading to an average life expectancy at birth of 65.5 years. HIV, Tuberculosis (TB), and malaria-related morbidity and mortality have significantly decreased in the last decades”.

Reviewer 2 

2. The manuscript is technically sound except for a few issues. In your costing approach, there was no mention of the possibility of double counting costs using the two approaches and how you handled double counting if it happened. If no double counting happened please mention it. 

Reply: This is an important point, and we fully agree when combining the result of the two methods there is a possibility of double counting of certain resources. However, in this particular analysis, we have ensured that these resources are cross-checked and accounted for at one location in order to avoid double counting. Therefore, we added the sentence in page number 7 and line number 131-132 “All cost inputs from both methods were crosschecked and accounted for to avoid double-counting”.

3. Please explain why the 2018 inflationary and exchange rate was used for costs data collected in 2019.

Reply: Thank you. We need to clarify that even though the primary data collection takes place between 7 June 2019 and 1 July 2019, real cost data has been collected for the year 2018 and before that date, which justifies the use of 2018 average annual exchange rate.

4. In table 2, please align supervision and its description properly.

Reply: Thank you. The cost inputs and its description is now aligned in Table 2.

5. Please indicate what software was used to make the linear interpolation between baseline and current coverage before the Spectrum analysis.

Reply: Thank you for your comment. We need to clarify that the spectrum software ((live saved tool -LiST, FamPlan, and TB impact and modelling estimate-TIME) is used to interpolate between the baseline and current coverage as indicated in page 9 line 170-174.

6. Personnel costs contribution for hygiene and environment has 3 percentage points but 4 items mentioned; utilization of latrines seems to have no percentage point mentioned

 Reply: We appreciate the reviewer’s comment and agree that we have missed one-percentage points. The sentences on page 14 and line number 263-266 have now been revised as “Personnel costs accounted for 18%, 24%, 24%, and 32% of the costs of improved water sources, hand washing with soap, the hygienic disposal of children’s stools, and the use of latrines, respectively. The remaining share of the expenditure was contributed by capacity building, including start-up and monitoring and evaluation.”

7. Major part of the discussion was comparing study results with other studies and little on impact of these results on current policy.

Reply: We appreciate the reviewer’s comments and agree that the discussion section needs to be strengthened by the impact of the result on the current policy. Thus, we have added the following two sentences on page 17 and line number 331-334.

The sentence now reads “The unit costs and cost-effectiveness of the HEP based on health need and supply constraints provide contextual evidence for decision-makers to prioritize healthcare interventions and allocate resources efficiently in order to improve population health.” And in page 18 and line number 364-366 “Thus, the HEP leads to a higher proportion of lives saved and a more cost-effective approach to delivering essential health services to rural and vulnerable communities, where access to qualified staff is limited”.

8. Small grammatical errors throughout the manuscript that need revision.

Reply: Thank you. We had attempted to correct the grammatical issues accordingly.

Response to comments from academic editor

Reply: Thank you for your comment. This manuscript has been revised according to the journal style requirements (please see the revised manuscript).

10. We note that you have indicated that data from this study are available upon request. PLOS only allows data to be available upon request if there are legal or ethical restrictions on sharing data publicly.

 Reply: Thank you for the suggestion. The data set used for this study has been uploaded as supporting information (in the revised manuscript).

11. Please amend your list of authors on the manuscript to ensure that each author is linked to an affiliation. Authors’ affiliations should reflect the institution where the work was done (if authors moved subsequently, you can also list the new affiliation stating “current affiliation:….” as necessary).

Reply: Thank you. We have amended the list of authors and ensured that each author is linked to an affiliation.

12. Thank you for stating the following in the Financial Disclosure section: 'Bill & Melinda Gates Foundation funded the study. The funder had no role in the conception, data collection, analysis and write up of this study.' We note that one or more of the authors are employed by a commercial company: Abenezer Consulting and Research PLC, and Monitoring, Evaluation, Research and Quality Improvement (MERQ) consultancy. 

Reply: Thank you for the clarification and suggestions. We have updated these sections accordingly. Please see pages 22 and 23, line number 471-478. 

13. Please also provide an updated Competing Interests Statement declaring this commercial affiliation along with any other relevant declarations relating to employment, consultancy, patents, products in development, or marketed products, etc. 

Please include both an updated Funding Statement and Competing Interests Statement in your cover letter. We will change the online submission form on your behalf. Please know it is PLOS ONE policy for corresponding authors to declare, on behalf of all authors, all potential competing interests for the purposes of transparency. PLOS defines a competing interest as anything that interferes with, or could reasonably be perceived as interfering with, the full and objective presentation, peer review, editorial decision-making, or publication of research or non-research articles submitted to one of the journals. Competing interests can be financial or non-financial, professional, or personal. Competing interests can arise in relationship to an organization or another person. Please follow this link to our website for more details on competing interests: http://journals.plos.org/plosone/s/competing-interests

Reply: Thank you for the clarification and suggestions. We have updated these sections accordingly. Please see page 22, line number 467-469. 

14. We note you have included a table to which you do not refer in the text of your manuscript. Please ensure that you refer to Table 2 and 4 in your text; if accepted, production will need this reference to link the reader to the Table.

Reply: We have now referred Table 2 and 4 in the main text of the revised manuscript.

---

## [Decision Letter · Decision Letter 1]

15 Jan 2021

Economic evaluation of Health Extension Program packages in Ethiopia

PONE-D-20-27778R1

Dear Dr. Assebe,

We’re pleased to inform you that your manuscript has been judged scientifically suitable for publication and will be formally accepted for publication once it meets all outstanding technical requirements.

Kind regards,

Athina Economou

Academic Editor

PLOS ONE

Additional Editor Comments (optional):

Reviewers' comments:

Reviewer's Responses to Questions

**Comments to the Author**

1. If the authors have adequately addressed your comments raised in a previous round of review and you feel that this manuscript is now acceptable for publication, you may indicate that here to bypass the “Comments to the Author” section, enter your conflict of interest statement in the “Confidential to Editor” section, and submit your "Accept" recommendation.

Reviewer #1: All comments have been addressed

Reviewer #2: All comments have been addressed

2. Is the manuscript technically sound, and do the data support the conclusions?

Reviewer #1: Yes

Reviewer #2: Yes

3. Has the statistical analysis been performed appropriately and rigorously? 

Reviewer #1: Yes

Reviewer #2: Yes

4. Have the authors made all data underlying the findings in their manuscript fully available?

Reviewer #1: Yes

Reviewer #2: Yes

5. Is the manuscript presented in an intelligible fashion and written in standard English?

Reviewer #1: Yes

Reviewer #2: Yes

6. Review Comments to the Author

Reviewer #1: (No Response)

Reviewer #2: All comments raised have been addressed appropriately, written in comprehensible English, and data made avialble for verification and confirmation of results of analysis done

7. PLOS authors have the option to publish the peer review history of their article (what does this mean?). If published, this will include your full peer review and any attached files.

Reviewer #1: **Yes: **Naomi Setshegetso

Reviewer #2: No

---

## [Editor Report · Acceptance letter]

19 Jan 2021

PONE-D-20-27778R1 

Economic evaluation of Health Extension Program packages in Ethiopia 

Dear Dr. Assebe:

I'm pleased to inform you that your manuscript has been deemed suitable for publication in PLOS ONE. Congratulations! Your manuscript is now with our production department. 

Kind regards, 

on behalf of

Dr. Athina Economou 

Academic Editor

PLOS ONE